# Diagnostic Tools for Cutaneous Leishmaniasis Caused by *Leishmania donovani*: A Narrative Review

**DOI:** 10.3390/diagnostics13182989

**Published:** 2023-09-18

**Authors:** Sachee Bhanu Piyasiri, Rajika Dewasurendra, Nilakshi Samaranayake, Nadira Karunaweera

**Affiliations:** Department of Parasitology, Faculty of Medicine, University of Colombo, Kynsey Road, Colombo 0800, Sri Lanka; sacheebhanu@gmail.com (S.B.P.); rajika@parasit.cmb.ac.lk (R.D.); nilakshi@parasit.cmb.ac.lk (N.S.)

**Keywords:** cutaneous leishmaniasis, *Leishmania donovani*, skin lesions, microscopy, molecular diagnosis, immunological diagnosis, histopathology, direct agglutination, diagnostic accuracy

## Abstract

Leishmaniasis, a neglected tropical disease, encompasses a spectrum of clinical conditions and poses a significant risk of infection to over one billion people worldwide. Visceral leishmaniasis (VL) in the Indian sub-continent (ISC), where the causative parasite is *Leishmania donovani*, is targeted for elimination by 2025, with some countries already reaching such targets. Other clinical phenotypes due to the same species could act as a reservoir of parasites and thus pose a challenge to successful control and elimination. Sri Lanka has consistently reported cutaneous leishmaniasis (CL) due to *L. donovani* as the primary disease presentation over several decades. Similar findings of atypical phenotypes of *L. donovani* have also been reported from several other countries/regions in the Old World. In this review, we discuss the applicability of different methods in diagnosing CL due to *L. donovani* and a comprehensive assessment of diagnostic methods spanning clinical, microscopic, molecular, and immunological approaches. By incorporating evidence from Sri Lanka and other regions on *L. donovani*-related CL, we thoroughly evaluate the accuracy, feasibility, and relevance of these diagnostic tools. We also discuss the challenges and complexities linked to diagnosing CL and review novel approaches and their applicability for detecting CL.

## 1. Introduction

Leishmaniasis is a vector-borne disease caused by protozoan parasites of the genus *Leishmania*. It constitutes a spectrum of clinical presentations with three main clinical entities recognized; viz. cutaneous leishmaniasis (CL), muco-cutaneous leishmaniasis (MCL), and visceral leishmaniasis (VL). CL is the most common, with an estimated annual incidence of 600,000 to 1 million new cases [1]. Typically, each *Leishmania* species is associated with a recognized clinical form in identified geographical setting(s). Old-World CL is caused by either *Leishmania tropica* or *L. major* while VL is caused by *L. donovani* (in the ISC, China, and Africa) and *L. infantum* (in the Mediterranean region) [2]. Post-kala-azar dermal leishmaniasis (PKDL), sometimes considered a fourth entity, is a sequel to VL caused by *L. donovani* and is reported in both the ISC and Africa [3,4,5,6,7,8,9,10,11,12,13,14,15,16,17]. In addition to the above, atypical “clinical form–*Leishmania* species” combinations are also reported in different parts of the world (Table 1). Sri Lanka was the first country to report such an atypical association of localized autochthonous CL in immunocompetent persons caused by *L. donovani*, a *Leishmania* species that usually causes VL [13]. Reports of similar atypical associations have since been reported from other foci as well [18,19,20,21,22] (Table 1). Most of these geographic areas from where atypical clinical forms are reported, tend to be known endemic sites of VL due to *L. donovani* or CL caused by other *Leishmania* sp. Some areas of India which were previously non-endemic for leishmaniasis were identified as endemic pockets for CL due to *L. donovani*, which is also a cause for concern [7,23].

Leishmaniasis is targeted for elimination in the ISC, with a focus on VL due to *L. donovani*. However, the early diagnosis and management of other phenotypes of this *Leishmania* species will also be of importance in eliminating parasite reservoirs which could contribute to persistent disease transmission [3]. Diagnosing CL caused by *L. donovani* is challenging due to diverse clinical presentations mimicking other skin conditions, leading to misdiagnosis and delayed treatment. Conventional methods such as microscopy can yield false-negative results, while limited awareness among healthcare providers hampers recognition and reporting. Variable parasite loads in lesions hinder detection, and interpretation of results requires expertise to avoid false positives or negatives. Diagnostic tests involving parasite material or those relying on the immune responses of the host could potentially exhibit inadequate performance due to the incompatibility of the parasite strains used. Adopting advanced diagnostics often poses a challenge due to resource constraints. Delays in diagnosis may allow for disease progression as well as its further spread. Addressing these issues is vital for improving the diagnosis and management of this form of leishmaniasis.

In a recent comprehensive review by de Vries et al., in 2022, the authors provide an in-depth exploration of the diagnosis of various forms of leishmaniasis, addressing current challenges and treatment options [24]. We describe the applicability of diagnostic tools used in conventional CL in diagnosing this atypical parasite species variant, with a focus on evidence generated through studies in Sri Lanka where the primary disease presentation due to *L. donovani* has continued to be CL for over three decades.

## 2. The Diagnostic Tests Recommended by World Health Organization for Cutaneous Leishmaniasis

The World Health Organization (WHO) [25] provides guidelines for various approaches to diagnosing CL. Recommended diagnostic methods encompass parasitological techniques, including microscopy to visualize *Leishmania* parasites in stained smears, culture for parasite growth and identification, and PCR for detecting *Leishmania* DNA in lesion samples. Serological methods, such as the direct agglutination test (DAT) and ELISA, are utilized to detect antibodies against *Leishmania* antigens. Molecular methods, such as PCR, enable the identification of *Leishmania* DNA in skin or blood samples, while immunological approaches, exemplified by the immunofluorescent antibody test (IFAT), to identify *Leishmania* antibodies using fluorescence microscopy (Figure 1).

The choice of method may vary based on resource availability, prevalent *Leishmania* species, and healthcare expertise. Additionally, commercial test kits such as CL Detect rapid diagnosis kits are available for efficient diagnosis of CL. However, the gold standard for diagnosing CL remains parasitological diagnosis achieved by directly visualizing *Leishmania* parasites in clinical samples using microscopy, known for its high specificity [24].

## 3. Clinical Identification

In dermatological conditions, the appearance of the lesions and their conformity to recognized patterns is central in reaching a probable diagnosis, and CL is no exception. The type of lesions found in Sri Lanka have been widely described [26,27,28,29,30,31]. These are single, dry lesions on exposed parts of the body, which start as a papule and enlarge to a nodule with central ulceration over a period of 1–6 months (Figure 2). A silvery scab/crust covering the ulcer, which is easily removed and then re-forms, is also described. A number of accompanying features of these usually non-tender, non-itchy lesions are also frequently observed, such as hypo/hyper-pigmentation, induration, and erythema of the surrounding skin. The clinical presentation of CL due to *L. donovani* bears resemblance to cutaneous lesions caused by other *Leishmania* species responsible for typical CL, such as *L. major*, *L. tropica*, *L. braziliensis*, *L. amazonensis*, and *L. mexicana*. However, while there are noticeable similarities, there also exist discernible differences in the clinical characteristics of skin lesions (Table 2) [32,33,34,35,36,37,38,39,40,41,42,43]. While the above presentation in a patient residing in an endemic region strongly supports a likely diagnosis of CL, differential diagnosis includes atypical mycobacterial infections, fungal infections, leprosy, sarcoidosis, ulcers due to other local causes such as venous stasis, and even skin neoplasms. No concomitant visceral involvement or past history of VL or progression of CL to VL is recorded in CL patients in Sri Lanka. In addition, different clinical appearances such as zosteriform, sporotrichoid, acne-form, erysipeloid, or plaque-type lesions, as well as lesions in unusual sites such as the ear lobe and buttocks, have also been documented [32,44,45] (Figure 2) (Table 2). Other foci which report CL due to *L. donovani* have also recorded similar clinical presentations [10,18,46,47,48].

Attempts have also been made to incorporate the clinical appearance and progression of the lesions into predictive scores aiding screening and diagnosis [49]. Overall, an acne-form, painless lesion progressing through the stages described above has been seen to have a comparatively high predictive value, while it is noteworthy that the sensitivity of most features in aiding a clinical diagnosis decreases as the lesion becomes more chronic. While self-resolving lesions have been reported [10], treatment of all patients with a parasitological diagnosis is recommended in Sri Lanka in view of the potential of the parasite species to cause systemic involvement through in vitro and ex vivo experiments [50]. Mucosal lesions, commonly involving the lips or the angle of the mouth, have also been reported in Sri Lanka as well as other foci reporting CL due to *L. donovani* [7,10,51]. The recorded number of VL due to *L. donovani* in Sri Lanka remains low [52] although there is a possibility that this condition may be under-diagnosed.

## 4. Direct Parasitological Methods for Diagnosis of CL

### 4.1. Direct Microscopy

The main methods include direct microscopic visualization of amastigotes in lesion aspirates (aspirating the lesion after injecting saline into it with a 23G needle), tissue scrapings (scraping the edge of the lesion using a sterile scalpel blade), impression smears using tissue biopsies, or isolation of parasites in culture in order to visualize the promastigote stage.

Direct smears are prepared by placing the aspirated lesion material or lesion tissue scraped from the edge of an active lesion on a glass slide or by gently rolling a 2–3 mm punch biopsy sample over a glass microscope slide to make an impression smear [53,54] (Table 3). The slides are fixed and stained with either Giemsa or hematoxylin–eosin stain. Microscopic examination under an oil immersion lens (X1000) enables the identification of *Leishmania* amastigotes confirming the diagnosis of CL [55].

Direct light microscopic methods exhibit a high level of specificity. These methods are characterized by their simplicity, speed, ease of performance, and cost-effectiveness. Some studies, including those by Wijesinghe et al., and Pulimood et al. [54,56] (Table 3), have documented the utility of direct microscopy in diagnosing CL caused by *L. donovani*.

The need for highly trained personnel to identify the presence of parasites in stained smears is a drawback. Furthermore, sensitivity might vary due to the method of sample collection, i.e., slit-skin smears have a higher sensitivity compared to lesion aspirates [57]. However, in a study conducted in the Southern province of Sri Lanka, the sensitivity of direct microscopy was much lower when compared to PCR and in vitro culturing methods [58]. Light microscopy does not distinguish between the species of *Leishmania* and, therefore, species diagnosis would require further investigation.

### 4.2. Histopathology

Histopathological diagnosis follows a standard procedure of obtaining a 2–3 mm punch biopsy from the active edge of the lesion and fixing the tissue in 10% formalin, followed by further processing where the tissue samples are dehydrated, cleared, embedded in paraffin (FFPE), cut into 4–5 μm thick sections and stained with hematoxylin and eosin. The amastigotes are visualized under a 100× oil immersion objective and typically seen inside tissue macrophages (histiocytes), while some may be also seen extra-cellularly.

Morphological alterations are seen in both the epidermis and dermis of the biopsy specimens [23,53,54,59,60]. Commonly reported epidermal changes include hyperkeratosis, acanthosis, parakeratosis and atrophy. Histiocytes and lymphocytes are common cell types seen in the dermal infiltrate. Varying degrees of granuloma formation have been reported in different settings, with higher parasite loads negatively associated with well-defined granulomas [59,61]. The spectrum of histological changes seen in CL patients ranging from diffuse infiltrates of parasitized macrophages to well-formed granulomas often resembles those reported for leprosy [62].

The histopathological assessment of CL provides a direct means of visualizing *Leishmania* parasites within tissue samples, thereby confirming infection. This approach not only precisely pinpoints the location of the parasites but also enables the visualization of the cellular immune response, thus contributing to the understanding of the underlying pathogenic mechanisms. The archival storage of histopathological samples supports future reference and serves as a valuable resource for research studies.

Despite these benefits, limitations include invasiveness, sampling errors, and the need for pathologists’ expertise. While histopathological identification is not a first-line test for the diagnosis of CL, it supplements clinical diagnosis, especially when dealing with lesions with doubtful morphological appearances.

### 4.3. Identification of Promastigotes by In Vitro Isolation (Parasite Cultures)

In vitro isolation of *Leishmania* sp. enables the definitive diagnosis of CL, which is important in proper clinical management. Culturing of *Leishmania* parasites, i.e., promastigotes, in suitable culture media has proven to be an effective method of diagnosis, with high sensitivity and specificity [10,17]. Culturing of parasites is also essential for further studies, such as parasite genomic studies, immune response evaluation, and vaccine candidate investigations.

The commonly used culture media include RPMI 1640, Locke’s semi-solid medium (LSSM), Novy–McNeal–Nicolle (NNN), and M199 [63]. The first successful attempt at in vitro culturing of the local *L. donovani* parasite was made using three standard media, i.e., the NNN medium, USAMRU (Difco blood agar), and Evan’s modified Tobie’s medium. Each medium was inoculated with saline aspirates from the lesions. Smears made of all culture media were positive between the fifth and ninth day of inoculation [64]. Similarly, an aspirate taken from the bone marrow of a suspected VL patient was successfully used for in vitro culturing of the parasites, indicating the usefulness of these culture methods in the diagnosis of either CL or VL [65].

Successful in vitro culturing and propagation of *Leishmania* parasites require technical expertise and sterile conditions, which are difficult to maintain in laboratories in peripheral areas [66]. A simple method for isolating and culturing *Leishmania* parasites using micro-capillary tubes was demonstrated by Allahverdiyev et al., in 2004 [67]. The same method was tried for the local *Leishmania* parasite *L. donovani* MON37 by Ihalamulla et al., in 2005 [68], when sterile saline aspirates from skin lesions were used to inoculate the culture medium (RPMI 1640 with added supplements) in micro-capillary tubes. This method is simple, cost effective, and economical compared to other traditional culture methods, since the volume of culture medium needed is less [68]. The inoculation of micro-capillary tubes has proven to be a highly sensitive method with a significantly lower risk of contamination when compared to the standard in vitro culturing techniques [68]. Furthermore, it could be transported (long distance) with ease without temperature control methods, which can be considered as an added advantage when collecting samples in the field [66].

### 4.4. Supplementary Tests to Histopathology

The detection of immune markers expressed by *Leishmania* using immunohistochemistry has been evaluated for the diagnosis of CL. Positive CD1a staining of amastigotes of Old World leishmaniasis by *L. major* and *L. tropica* has been previously reported [69]. In 2023, Riyal et al., identified a similar staining pattern in *L. donovani* causing CL for the first time, with a sensitivity of 74% [61]. While fluorescence in situ hybridization (FISH) is not a common approach in diagnosing CL, applying *Leishmania* genus-specific FISH probes on FFPE tissue has been shown to improve sensitivity over conventional staining (Table 3) [70].

## 5. Molecular Techniques for Identifying *Leishmania* Parasites

Molecular methods of detecting parasites in biological samples are considered superior for species identification of the causative agent of leishmaniasis due to their higher sensitivity and specificity. They may be routinely used for diagnosis in laboratories with the required facilities and resources. These methods are considered important in places where several species of *Leishmania* are responsible for the disease, e.g., in Brazil, where at least seven *Leishmania* sp. are responsible for causing CL [55].

### 5.1. Polymerase Chain Reaction (PCR)

Several PCR-based protocols have been developed locally, to detect the local *L. donovani* parasite (Table 3). PCR does not require abundant test material (i.e., cultured parasites) and the clinical samples obtained from patients could be directly subjected to PCR. This can be considered as a significant advantage of PCR-based methods.

*Leishmania* kinetoplast DNA contains small circular molecules called kDNA mini-circles. These are ideal targets for sensitive *Leishmania* detection due to their high copy numbers and conserved sequences. Despite challenges posed by diverse mini-circle networks, strain typing using this genomic region is complicated. Previously, characterizing *Leishmania* mini-circulomes required isolating and cloning before sequencing. A study by Kocher et al., in 2018 demonstrated that high-throughput sequencing of individual mini-circle PCR products bypasses these labor-intensive laboratory steps [71]. Ranasinghe et al., in 2015 [72], demonstrated a modified PCR protocol (two *Leishmania*-genus-specific and one *L. donovani*-species-specific PCR assays) for the identification of *Leishmania* sp. in 38 skin biopsy samples taken from local CL patients (Table 3). The genus-specific PCR assays showed >92% sensitivity, whereas the species-specific PCR assay only showed a sensitivity level of 71.1% (Table 3). Similarly, the identification of CL was achieved using the *Leishmania* kinetoplast DNA sequence in a recent study conducted by Lata et al., in 2021, in India [7]. Furthermore, a modified nested PCR method tested for *L. donovani* spp. with 100% sensitivity for 40 true positive and 30 true negative samples has also been described [73] (Table 4).

### 5.2. PCR–RFLP

PCR–RFLP is a molecular technique that involves digesting PCR-amplified products with restriction enzymes, generating unique polymorphic fragments that serve as markers for identifying *Leishmania* species [74]. PCR–RFLP methods have been used successfully to identify the causative agent of leishmaniasis up to species level, specifically using internal transcribed spacers (ITSs). ITSs are non-coding regions of rRNA genes which are known to be highly conserved among the *Leishmania* sp. That includes *L. donovani*, *L. infantum*, *L. major*, *L. tropica* and *L. aethiopica*. Hence, PCR amplification of the ITS region followed by restriction digestion by *Hae III* can be considered as a suitable method for identification and differentiation between all medically important *Leishmania* parasite species [75,76,77] (Table 4).

A modified method of this nested PCR was developed and tested on samples from 194 CL patients, and it proved to be successful with patients with atypical lesions and with low parasite loads (Table 4) [75]. In India, a case was identified through the examination of both blood and skin samples using polymerase chain reaction (PCR). By employing restriction fragment length polymorphism (RFLP), the *Leishmania* parasite was successfully identified and classified as *L. donovani*. This investigative approach facilitated the confirmation of concurrent infections of both visceral leishmaniasis and dermal leishmaniasis within the same individual [78].

### 5.3. LAMP

Loop-mediated isothermal amplification (LAMP) is a method that amplifies DNA rapidly under isothermal conditions with high specificity and efficiency [79].

The LAMP assay was tested as a potential diagnostic tool to diagnose CL patients in Sri Lanka (Table 4) [79]. This study reported 82.6% sensitivity and 100% specificity with positive and negative predictive values of 100% and 66%, respectively, compared to microscopy as the gold standard. This proved to be a more efficient and cost-effective method, with an assay procedure that can be completed in 1 h and 40 min, compared to nested PCR which takes approximately 3 h and 30 min [80].

### 5.4. Recombinase Polymerase Amplification Assay (RPA)

Gunaratna et al., in 2018 [81], introduced a recombinase polymerase amplification-assay-based mobile suitcase laboratory, as a point-of-care diagnostic method for CL. This was tested with different sample types (i.e., punch biopsy, slit-skin smear and fine-needle aspirate) for optimal performance. The method was performed by the rapid extraction of DNA using SpeedXtract (Qiagen, Hilden, Germany) from patient samples and subjecting it to RPA assay using RPA primers specific to the kinetoplast mini-circle DNA of *L. donovani* (Table 3). The presence of parasite DNA was detected by measuring fluorescence intensities with a portable fluorescence reader (T8 Axxin, Fairfield, Australia). RPA had a 65.5% sensitivity under field conditions, which was close to but higher than the sensitivity of PCR (63.4%) conducted on biopsy samples preserved in RNA and later transported to the processing laboratory. The speed at which the results could be obtained (within 35 min) by RPA, when compared to conventional molecular methods of *Leishmania* detection (i.e., PCR, 8 h of test performance) was highlighted as the main advantage [81], though the cost of the product has limited its use in the field.

Techniques such as RPA and LAMP provide a way to obtain relatively quick results without the necessity for extensively skilled professionals.

### 5.5. Multi-Locus Enzyme Electrophoresis (MLEE)

The differences in the migration patterns of enzymes on a gel reflect the differences in single nucleotide polymorphisms (SNPs) in the genes encoding these enzymes and can be detected via gel electrophoresis. This is the basis of MLEE to identify SNPs in the coding genes [82]. Isoenzyme typing using MLEE was performed to identify the *Leishmania* species from an autochthonous VL patient in a study by Ranasinghe et al., 2012 (Table 4) [15].

### 5.6. Multi-Locus Microsatellite Typing (MLMT)

Isoenzyme typing of 15 different enzymes in two reference *L. donovani* strains (i.e., *L. donovani* MON-2 and MON-37) together with the suspected VL sample was performed using a method described by Alam in 2009 [83] (Table 4). MLMT was employed to conduct a comparative analysis of *L. donovani* strains that fall under the MON-37 zymodeme. This comparison involved strains originating from Cyprus and Israel, as well as strains from the Indian subcontinent, the Middle East, China, East Africa, and strains belonging to other zymodemes [83]. The genetic analysis of the MON-37 strains revealed that those from Kenya, Sri Lanka, and India were more genetically similar to strains of other zymodemes from their respective regions than to MON-37 strains from different geographical areas. In contrast, MON-37 strains from Cyprus and Israel were distinct not only from each other but also from all the other MON-37 strains studied. This suggests that the Cyprus and Israel strains are likely indigenous to their respective regions [83].

**Table 4 diagnostics-13-02989-t004:** Modified PCR methods for the detection of cutaneous leishmaniasis due to *Leishmania donovani*.

*Methods*	Species Determination	Primers	References
** *PCR based on KDNA amplification* **	1. kDNA *Leishmania*-genus specific 2. kDNA *Leishmania donovani*-species-specific PCR.	**1. JW11 (forward):** 5′-CCTATTTTACACCAACCCCCAGT-3′ **JW12 (reverse):** 5′-GGGTAGGGGCGTTCTGCGAAA-3′ **2. LdF (forward):** 5′-AAATCGGCTCCGAGGCGGGAAAC-3′ **LdR (reverse):** 5′-GGTACACTCTATCAGTAGCAC-3′	[7,72]
** *Nested PCR* **	*Leishmania* genus-specific primers	**outer primers** **P221:** 5′-GGTTCCTTTCCTGATTTACG-3′ **P332:** 5′-GGCCGGTAAAGGCCGAATAG-3′ **inner primers** **P223:**	[73]
** *ITS1 PCR amplification followed by RFLP* **	*Leishmania*-genus-specific	**Outer primers** **LITSR** 5′-CTGGATCATTTTCCGATG-3′ **L5.8S** 5′-TGATACCACTTATCGCACTT-3′ **Inner primers** **LITSR inner** 5′-CATTTTCCGATGATTACACC-3′ **L5.8S inner** 5′-TACTGCGTTCTTCAACGA-3′	[74,75,76]
** *LAMP for kinetoplast minicircle DNA* **	*Leishmania*-genus-specific	**Primers for first round** **R 221:** 5′GGTTCCTTTCCTGATTTACG3′ **R332:** 5′GGCCGGTAAAGGCCGAATA3′ **Primers for the nested PCR** **R223:** 5′TCCCATCGAACCTCGGTT3′ **R333:** 5′AAAGCGGGCGCGGTGCTG3′	[80]
** *Recombinase Polymerase Amplification Assay (RPA)* **	*Leishmania*-genus-specific	**RPA primers** FP3: 5’-ATG GGC CAA AAA CCC AAA CTT TTC TGG TCC TC-3’ **RP3:** 5’-CTC CAC CCGACC CTA TTT TAC ACC AAC CCC CAG T-3’ **Probe:** CGC CTC GGA GCC GAT (BHQ1dT) (Tetrahydrofuran) (FAMdT) TGG CAT TTT TGG CTATTT TTT GAA CGG GAT-phosphate)	[81]
** *Multi-locus Enzyme Electrophoresis (MLEE)* **	*Leishmania donovani* MON-37 zymodeme	**6PGDH-Forward:** AATCGAGCAGCTCAAGGAAG **6PGDH-Reverse:** GAGCTTGGCGAGAATCTGAC)	[15]
** *Multi-locus Microsatellite Typing (MLMT)* **	*Leishmania donovani* MON-37 zymodeme	The 14 variable microsatellite markers Li 22-35, Li 23-41, Li 41-56, Li 45-24, Li 46-67, Li 71-5/2, Li 71-7, Li 71-33, Lm2TG, Lm4TA, TubCA, CS20, kLIST 7031 and kLIST 7039 were used	[83]

## 6. Immunological Techniques

Serological methods for detecting leishmaniasis infection, which include enzyme-linked immunosorbent assay (ELISA), immunochromatographic strip test (ICT), indirect fluorescent antibody test (IFAT), Western blot, direct agglutination test (DAT), and latex agglutination test (KAtex) primarily concentrate on identifying anti-leishmanial antibodies in blood, urine, or saliva [83,84]. These methods are commonly used for the serological diagnosis of VL due to the presence of anti-leishmanial antibodies, but they are not widely used in CL diagnosis due to poor humoral response and low sensitivity [85].

### 6.1. Enzyme-Linked Immunosorbent Assay (ELISA) Based Diagnostics

Highly sensitive in-house ELISA tests have been developed for the detection of CL, leading to improved sensitivity and specificity. Hartzell et al., in 2008 [86], conducted a study on soldiers with CL using a rK39 dipstick and the study revealed that some patients without clinical evidence of VL had a reactive rK39 assay result. This finding suggests a potential association between CL and the positivity of rK39 assay. On the other hand, a study by Svobodova et al., in 2009 [87], revealed all subjects testing negative in the rK39 assay for the transmission of an atypical form of CL caused by *L. infantum* in South Anatolia (Table 5).

In the Sri Lankan context, De Siva et al., 2022 [88], focused on an endemic hotspot of CL and developed an in-house ELISA using the recombinant kinesin-related protein (KRP42 antigen), which is a homolog of rK39, to detect CL [89]. The assay was compared to ITS-1 nested PCR and showed a sensitivity and specificity of 94.4% for serum samples, and a sensitivity and specificity of 61.7% and 66.8%, respectively, for urine samples (Table 5).

Another successful approach has been the use of crude antigens derived from amastigotes of local *Leishmania* strains [88]. Seroprevalence studies by Siriwardana et al., 2018, and Deepachandi et al., 2020 [90,91], demonstrated that in-house ELISA assays with soluble crude *Leishmania* antigens (SLAs) can identify active CL infections, with a sensitivity level of 82.0%. An indirect ELISA assay with SLAs developed by Piyasiri et al., in 2022 [92], showed high performance, with a sensitivity of 98% and specificity of 90.3%. This assay served as a tool not only for diagnosing CL patients with active disease but also for detecting exposure to CL in endemic individuals within the local setting. The study also compared and contrasted the humoral response driven by anti-leishmanial IgG antibodies in CL patients and endemic individuals.

### 6.2. Immunochromatographic Strip Test (ICT)

rK39 antigen-based immunochromatographic strips (Table 5), widely used for detecting VL infections, have been tested for their applicability in detecting CL infections. A study in Brazil using the commercial rK39 strip test (Kalazar Detect^TM^ by In Bios International, Inc., Seattle, WA, USA) confirmed the absence of serological cross-reactions between individuals with CL and those with VL caused by *L. infantum* [93]. However, the same strip test was not found to be sensitive in detecting CL cases in Sri Lanka [50]. Furthermore, Ejazi et al., in 2019 [94], used a dipstick against *L. donovani* membrane antigens (Lag), which showed low sensitivity (60%) for detecting CL infection in Sri Lanka, indicating that rK39 is not a suitable antigen candidate for CL detection in Sri Lanka, which might be due to the low immunogenicity of *L. donovani* rK39-based antigens in CL and also the low sensitivity of the rK39 strip test in the detection of low levels of generated antibodies in CL patients.

The peroxidoxin antigen (Table 5) has been recently used in ELISA-based diagnostic assays, including the CL Detect™ rapid test by Inbios International, USA. The CL Detect™ rapid test showed low sensitivity (31.3%) in its detection of *L. donovani* when tested on skin slits and dental broach samples in a study conducted in Ethiopia [95]. Similarly, peroxidoxin-based diagnostic test kits were found to have low sensitivity to *L. major* and *L. tropica* as they require specific samples from skin ulcers, according to a 2018 study by Bennis et al. [96]. Furthermore, the CL Detect™ IC-RDT, originally designed to detect *L. major*, demonstrated poor sensitivity for diagnosing CL infections due to *L. donovani* in Sri Lanka [50,97].

Currently, there is no published evidence of promising serology-based rapid diagnostic test kits for *L. donovani*-induced CL. However, preliminary findings using the KMP11 antigen suggests it to be a promising candidate in diagnosing active CL infection in Sri Lanka and studies are currently underway to determine its potential as a diagnostic tool (Karunarathilaka et al., unpublished data).

### 6.3. Direct Agglutination Test (DAT)

The direct agglutination test (DAT) has been widely used for more than 25 years as a diagnostic tool for VL and to some extent for the typical form of CL caused by *L. major*, *L. tropica*, *L. mexicana*, *L. braziliensis and L. amazonensis*. It has been found to have high clinical accuracy, with the accuracy of diagnosis depending on the titer of antibodies in the serum [97]. The presence of anti-*Leishmania* antibodies in the tested serum leads to the formation of a pale blue film over the well, indicating a positive result [98]. DAT can also be a useful addition to the diagnosis of CL due to *L. donovani* from serum samples [99].

**Table 5 diagnostics-13-02989-t005:** Immunological methods for the diagnosis of cutaneous leishmaniasis.

Methods	Target Antigen	Commercial Product/In-House Assay	References
**Enzyme linked immunosorbent assay (ELISA)**	rK39	In-house assay	[26,86,87]
	KRP42	In-house assay	[88]
	Crude soluble *Leishmania donovani* antigen	In-house assay	[89,90,91,92]
**Immunochromatographic strip test (ICT)**	rK39	Kalazar Detect^TM^ by In Bios International, Inc., Seattle, WA, USA	[93,94]
	Peroxidoxin	CL Detect^TM^ by In Bios International, Inc., Seattle, WA, USA	[50,95,97]
**Direct agglutination test (DAT)**	Crude parasite antigen	In-house assay	[99]

## 7. Challenges in Diagnosing Atypical Cutaneous Leishmaniasis Caused by *Leishmania donovani*

Distinguishing CL from other skin conditions poses a significant diagnostic challenge [31]. The clinical spectrum of atypical presentations often mirrors various dermatological disorders, making accurate differentiation complex. Lesions may resemble eczema, psoriasis, fungal infections, or bacterial cellulitis, among others. This overlapping clinical manifestation can lead to misdiagnosis and subsequent delays in appropriate treatment [44]. Thus, a comprehensive understanding of the distinct features and a thorough evaluation of the clinical, microscopic, molecular, and immunological aspects of CL are essential for achieving precise diagnoses and facilitating effective management strategies. To circumvent this issue, healthcare providers should prioritize a comprehensive approach that combines clinical assessment, appropriate laboratory testing, and consideration of the patient’s travel history and the endemicity of the disease in the region.

## 8. Innovative Approaches for Diagnosis of Cutaneous Leishmaniasis

Exploring accurate novel techniques alongside conventional methods is imperative to detect CL effectively.

Saavendra et al.’s 2020 [100] study in Peru revealed that high-frequency ultrasound offered a promising approach for the non-invasive visualization of *Leishmania (Viannia) braziliensis*-induced CL. Their findings demonstrated a strong correlation between ultrasound findings and histopathological CL characteristics, suggesting that high-frequency ultrasound could be a reliable diagnostic tool for CL in resource-limited settings.

Furthermore, an artificial intelligence (AI)-based algorithm is currently utilized for the automated detection and diagnosis of leishmaniasis. In 2022, Zare et al. [101] devised an algorithm for *Leishmania* parasite detection using integral image representation, facilitating faster processing. The study achieved a recall rate of 65% and a precision rate of 50% for detecting leishmania-infected macrophages. This tool’s versatility extends to identifying unusual patterns of atypical CL skin lesions. Khatami et al. [102] in Iran examined the biophysical and ultrasonographic properties of CL lesions and found correlations with histological features. A proof-of-concept study conducted in Sri Lanka utilizing photographic imaging has showcased the promising potential of novel methods, particularly for predicting lesion response to anti-leishmanial treatment. It is important to note that while this innovative approach has not yet been implemented in routine clinical practice in Sri Lanka, its demonstrated potential suggests it could be a valuable addition to the arsenal of tools for managing CL in the future [103]. Moreover, establishing a comprehensive image library of atypical CL lesions would further enhance their practical applications.

Clustered regularly interspaced short palindromic repeats/CRISPR-associated protein (CRISPR/Cas) systems are advanced tools for nucleic acid detection, offering high specificity, sensitivity, and speed. This technology is considered an ideal point-of-care test and it is versatile for various applications in the detection of CL in some geographical regions. In 2022, Duenas et al. [104] employed CRISPR-Cas12a for detecting *Leishmania* spp. in Peru and other Latin American endemic areas, with a particular emphasis on *Leishmania (Viannia) braziliensis*, the primary agent responsible for CL and MCL. However, with limited technology, this diagnostic method may not be feasible in local settings.

Studies conducted by Castillo-Castaned et al., in 2022 and Patino et al., in 2021 yield significant findings in several domains [105,106]. They demonstrate the effective identification of trypanosomatid co-infections within clinical samples of *Leishmania*, a crucial discovery given the potential complications posed by co-infections with pathogens like leprosy or HIV. This enables healthcare providers to tailor treatment strategies to address multiple health issues in affected individuals. The application of next-generation sequencing techniques, focusing on the heat shock protein 70 gene, enhances the precision of genetic analysis, enabling the detection of subtle variations and co-infections that might be elusive through conventional diagnostic methods. These studies notably center on atypical cases of CL, uncovering instances where CL co-occurs with conditions such as leprosy or HIV, thus facilitating the early diagnosis and appropriate management of these complex cases. These findings have wide-ranging biological, clinical, and epidemiological implications, informing public health strategies, treatment guidelines, and surveillance efforts, ultimately contributing to improved disease control and management.

Buffi et al.’s 2023 study [107] presents a groundbreaking approach to improving leishmaniasis diagnostics with profound implications. Their utilization of high-resolution melting (HRM) analysis to pinpoint informative polymorphisms in single-copy genes encoding metabolic enzymes represents a significant leap forward, offering highly accurate and species-specific insights into *Leishmania* parasites, especially *L. infantum*. This precision is crucial for tailoring treatment strategies. Furthermore, the development of rapid genotyping assays based on HRM simplifies the genotyping process, replacing labor-intensive and specialized methods like multi-locus enzyme electrophoresis (MLEE) and multi-locus microsatellite typing (MLMT). These innovations hold promise not only in the efficient identification of *L. infantum* but also in the potential transformation of atypical CL detection, enabling earlier and more accurate diagnoses with broader accessibility and cost-effectiveness, ultimately benefiting both clinicians and patients, particularly in endemic regions.

These findings hold potential for the non-invasive early detection of CL and as methods of outcome measures in clinical trials for evaluating new treatment modalities. Moreover, these approaches significantly enhance the diagnostic accuracy of CL.

## 9. Conclusions

Diagnostic options for atypical *L. donovani*-induced CL closely resemble those available for conventional CL. Direct microscopy of lesion material is the most cost-effective option which, along with the minimally invasive nature of sampling involved, makes it the method of choice to be implemented in resource-poor settings. Molecular methods (i.e., MLEE, RPA, NGS, LAMP), on the other hand, offer heightened sensitivity, crucial for detecting low parasite numbers in chronic or partially treated lesions and will help to correctly identify the *Leishmania* species. Point-of-care tests are the need of the day and the identification of candidate antigens with high immunogenicity combined with high sensitivity and specificity to be incorporated in rapid assays should be a priority. Moreover, advancing these assays to differentiate active disease from exposure for disease surveillance in endemic regions should also be explored. It is important to acknowledge the challenges associated with detecting atypical CL due to the differences in local *L. donovani* parasites.

## Figures and Tables

**Figure 1 diagnostics-13-02989-f001:**
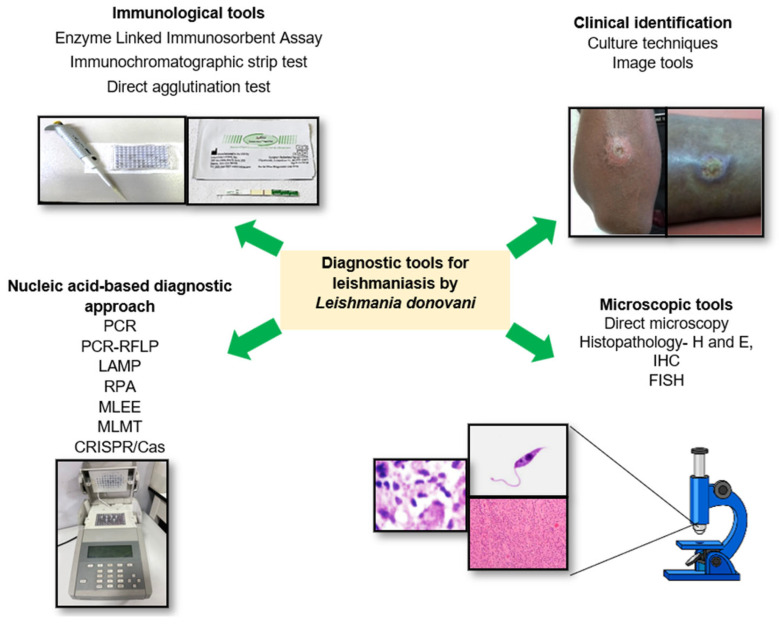
Graphical representation of diagnostic methods of cutaneous leishmaniasis. PCR: polymerase chain reaction, RFLP: restriction fragment length polymorphism, LAMP: loop-mediated isothermal amplification. RPA: recombinase polymerase amplification assay. MLEE: multi-locus enzyme electrophoresis. MLMT: multi-locus microsatellite typing.

**Figure 2 diagnostics-13-02989-f002:**
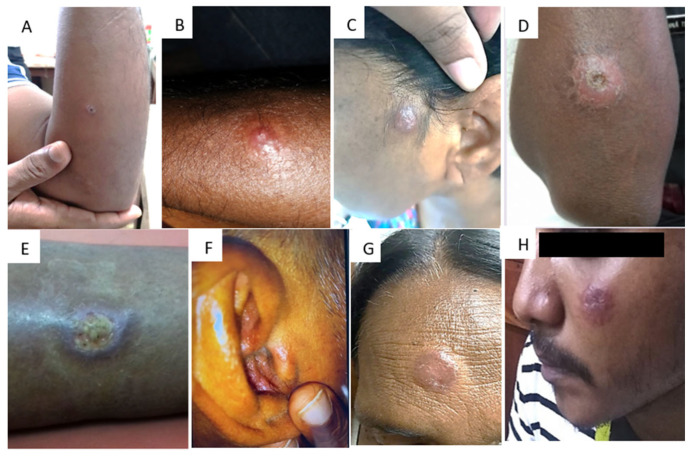
Spectrum of cutaneous lesions caused by *Leishmania donovani*. Typical appearance of lesions: (**A**) papule, (**B**,**C**) nodule, (**D**) ulcerative nodule, (**E**) ulcerative plaque. Atypical appearance of lesions: (**F**–**H**).

**Table 1 diagnostics-13-02989-t001:** Geographical distribution of cutaneous leishmaniasis due to *Leishmania donovani*.

Country/Region	Typical CL Caused by	Atypical CL Caused by	Reference
Cyprus	*L. major* *L. tropica*	*L. donovani* *L. infantum*	[5,18,19]
Ethiopia	*L. aethiopica*, *L. major*, *L. tropica*	*L. donovani*	[4,6]
India (Himachal Pradesh and Kerala)	*L. major*, *L. tropica*	*L. donovani*	[7,8,9,10,23]
Israel	*L. major*, *L. tropica*	*L. donovani*, *L. infantum*,	[4,5,20]
Kenya	*L. tropica*, *L. aethiopica*, *L. major*	*L. donovani* zymodeme z6	[11,12]
Sri Lanka	No typical form of CL is existing	*L. donovani* Mon 37 zymodeme	[5,13,14,15,16]
Sudan	*L. major*	*L. donovani* Mon-82 zymodeme	[17]
Uganda	No typical form of CL is existing	*L. donovani*	[4,5]
Yemen	*L. major* *L. tropica*	*L. donovani*	[21,22]

**Table 2 diagnostics-13-02989-t002:** Comparison of cutaneous lesions caused by different species of *Leishmania*.

Species Causing Typical CL	Clinical Features of CL Lesions		References
	Similarities to Lesions Caused by *L. donovani*	Differences to Atypical Lesions Caused by *L. donovani*	
*L. major*	Dry, crusted, and scaly lesions. Localized to the site of the sand fly bite. Lesions have a central depression and a raised border. Painless, with minimal inflammation.	Wet lesions are typically larger, more inflamed. Lesions have an ulcerative appearance with moist, weeping, or pustular areas. Painful.	[32,33,34,35]
*L. tropica*	Typically, dry, ulcerative sores with a scaly border. Localized to the site of the sand fly bite.	Wet lesions appear larger, more inflamed, and exudative. Painful and more uncomfortable.	[32,33,36,37]
*L. braziliensis*	No similarities shown.	Typically ulcerative in nature, with a central area of ulceration and raised, inflamed borders. Vary in size and shape, from small ulcers to larger sores.	[33,38,39,40,41]
*L. amazonensis*	Non-ulcerative nodules or plaques on the skin. Lesions can be dry, scaly, and may have a raised border. Painless.	Present as multiple lesions.	[33,41]
*L. mexicana*	Localized lesions. Lesions often start as a single nodule or ulcer at the site of the sand fly bite. The lesions may be painless or only mildly painful.	A distinctive form of localized cutaneous lesion involving the ear known as a “chiclero ulcer”.	[33,42,43]

**Table 3 diagnostics-13-02989-t003:** Microscopic methods for diagnosis of cutaneous leishmaniasis due to *Leishmania donovani*.

Method	Tissue/Sample Used	Microscopy	Reference
*Giemsa staining*	lesion aspirates scrapings/slit-skin smears	Direct microscopy—light microscope	[28,31,33]
*Hematoxylin-Eosin staining*	lesion aspirates scrapings/slit-skin smears	Direct microscopy—light microscope	[29,31]
*Fluorescence in situ Hybridization method (FISH*)	scrapings/slit-skin smears tissue biopsies	Fluorescent microscope	[45]
*Histopathological methods*	tissue biopsies or isoleted parasites	Direct microscopy—light microscope	[23,28,29,34,35,37]

## Data Availability

No new data were created or analyzed in this study. Data sharing is not applicable to this article.

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
