# Peer review of "Diagnostic Tools for Cutaneous Leishmaniasis Caused by Leishmania donovani: A Narrative Review"

_diagnostics, 2023, doi:10.3390/diagnostics13182989_

Round 1

Reviewer 1 Report

The authors have written a review article entitled “Diagnostic tools for cutaneous leishmaniasis caused by Leishmania donovani: a narrative review”. There is a recently published similar kind of work, de Vries, H.J.C., Schallig, H.D. Cutaneous Leishmaniasis: A 2022 Updated Narrative Review into Diagnosis and Management Developments. Am J Clin Dermatol 23, 823–840 (2022). The authors should cite this work and provide more updated information related to diagnosing atypical cutaneous leishmaniasis caused by Leishmania donovani, especially from Asian perspectives. I have the following questions about the present form of this manuscript. If you can include those and make it in a concise form, it would be really of interest to the readers who are working in Leishmania donovani causing cutaneous leishmaniasis

Ø  Abstract- The present attempt is to discuss on various diagnostic tools for Leishmania donovani. However, there is nothing related to this in the abstract. The authors should rewrite the abstract and keywords.

Ø  In the introduction, the authors should highlight, the problems or issues present in the diagnosis of atypical cutaneous leishmaniasis caused by Leishmania donovani. What is unique or is there any deviation from the routine diagnosis of cutaneous leishmaniasis?

Ø  Table 1 is highly confusing. I suggest the authors enlist various cutaneous leishmaniosis events caused by Leishmania donovani only in various parts of the world.

Ø  In the clinical presentation section, discuss the differences between typical cutaneous leishmaniosis and atypical ones with respect to clinical manifestations, hallmark signs, etc.

Ø  Before elaborating on various diagnostic approaches, describe the tests recommended by the WHO. And indicate gold standard tests, any commercial kits available, etc.

Ø  Include a figure classifying various diagnostic approaches and include the tests under the heading of broader approaches. For eg. Nucleic acid-based diagnostic approach (antigen detection)- PCR, LAMP, RPA, etc.

Ø  Under each diagnostic approach include a table furnishing various tests available, their sensitivity, specificity, detection limits, disadvantages, etc.

Ø  Include a table highlighting the commercial tests routinely used for the diagnosis of atypical cutaneous leishmaniasis.

Ø  Introduce a paragraph on the differential diagnosis of atypical cutaneous leishmaniosis and how to circumvent the misdiagnosis?

Ø  Introduce a paragraph on recent advances in the diagnosis of atypical cutaneous leishmaniasis.

Ø  Conclusions- Should be completely rewritten. Based on your literature survey, suggest some diagnostic approaches for accurate diagnosis of atypical cutaneous leishmaniasis.

Ø  I expect a better useful version of the current manuscript so that the readers will be benefitted.

Ø  Extensive editing of English is required.

Extensive editing of English is required.

Reviewer 2 Report

The review is well written and worthy for the community working on leishmaniasis. Below please find some comments for improvement of the manuscript.

l   Line 31: Italicize L. infantum

l   Line 40: Italicize Leishmania

l   Table 1: Thee Sub species/strain name is left blank in the raw of ‘Ethiopia’. Fill it.

l   Table 1: Rather than citing Reference 5 (WHO report), it is more proper to cite original articles describing the atypical cases in each region.

l   Line 138: Delete ‘in’

l   Line 140: includes to include

l   Line 149 ‘In’ to ‘in’, and italicize ‘in vitro’

l   Line 189: My understanding is that kDNA is rather less conserved compared with the other PCR targets on the chromosomes. Can you provide any references describing almighty of kDNA PCR in detecting a wide range of Leishmania species?

l   Line 193: Reference 46 is not an appropriate paper for claiming the usefulness of kDNA PCR for leishmaniasis as it tests only a single species, i.e., L. mexicana.

l   Line 206: There should be even better references for PCR-RFLP than Reference 49 which is about Fusarium.

l   Table 3. On the top row, I believe ITS1 is not kDNA.

l   Line 266: ‘Serological methods for detecting Leishmania’ is misleading as some of the methods described detect not the parasites but indirect evidence of Leishmania infection. Please rephrase.

l   Table 4 is missing.

l   Line 538: The journal name for Reference 46 seems to be wrong. Similar errors exist in the Reference section, so please check.

Reviewer 3 Report

Leishmania, vectored by female sandflies, is rampant in tropical and Mediterranean climate with estimated occurrence of upto a million people annually. To eliminate or control this disease, cost effective but robust diagnostics need to be identified and applied. Authors Piyasiri et al review the current types, clinical and atypical manifestations of atypical cutaneous leishmaniasis (CL) with particular focus on diagnostics currently in use in Srilanka. The causal agent of such atypical leishmanisasis is L. donovani, an agent typically associated with visceral leishmaniasis- suggesting potential difficulties in correct diagnosis and treatment.

The review satisfactory in covering most diagnostic approaches. Cost effective techniques such as direct microscopic visualization of amastigotes in lesions, histopathology or in-vitro isolation comprise an established spectrum of diagnosis. However, the requirement for skilled and trained technical experts to both prepare and correctly identify patients’ samples may hinder the overall effectiveness of these methods. At the same time, the methods are largely examination dependent, and can be mis-diagnosed based on the person performing the task. In contrast, molecular methods such as PCR (kDNA amplification, ITS gene amplification followed by restriction digestion), RPA and LAMP (Loop-mediated isothermal amplification) and largely user-agnostic. These provide a method to provide comparatively rapid results without the need for highly specialized technicians. However, the initial equipment and training cost may be a hindrance in its wide applicability, particularly in areas with low economic resources.

Although the article covers various established diagnostics for CL detection, it fails to mention some of the more interesting upcoming diagnostic features.  Despite mentioning CRISPR/Cas system and potential application along with other novel diagnostic tools, the review completely disregards the genre of next generation sequencing (NGS). Articles detailing the detection of patient samples with co-infections involving multiple leishmania strains using both Sanger and NGS sequencing (Castillo-Castaneda et al, 2022, Patino et al 2021) should be included here. Similarly, Buffi et al (2023) detail the use the HRM assay (High resolution melt curve analysis) for Leishmania genotyping which, if established, can be a revolutionary way to change diagnostics as it is cost effective and requires very little in terms of specialized personnel.

Overall, this article could be improved by including more novel diagnostic approaches and providing the scope for implementation of such techniques for rapid diagnostics impacting detection and treatment of Leishmaniasis.

Round 2

Reviewer 1 Report

The authors have addressed my comments

Minor editing of English language required